# T cell-intrinsic role of IL-6 signaling in primary and memory responses

Simone A Nish[1†], Dominik Schenten[1†], F Thomas Wunderlich[2†], Scott D Pope[1], Yan Gao[1], Namiko Hoshi[1], Shuang Yu[1], Xiting Yan[3], Heung Kyu Lee[1], Lesley Pasman[1], Igor Brodsky[1], Brian Yordy[1], Hongyu Zhao[3], Jens Brüning[2], Ruslan Medzhitov[1,4]*

[1]Department of Immunobiology, Yale University School of Medicine, New Haven, United States; [2]Max Planck Institute for Neurological Research, Cologne, Germany; [3]Department of Biostatistics, Yale School of Public Health, New Haven, United States; [4]Howard Hughes Medical Institute, Yale University School of Medicine, New Haven, United States

**Abstract** Innate immune recognition is critical for the induction of adaptive immune responses; however the underlying mechanisms remain incompletely understood. In this study, we demonstrate that T cell-specific deletion of the IL-6 receptor α chain (IL-6Rα) results in impaired Th1 and Th17 T cell responses in vivo, and a defect in Tfh function. Depletion of Tregs in these mice rescued the Th1 but not the Th17 response. Our data suggest that IL-6 signaling in effector T cells is required to overcome Treg-mediated suppression in vivo. We show that IL-6 cooperates with IL-1β to block the suppressive effect of Tregs on CD4+ T cells, at least in part by controlling their responsiveness to IL-2. In addition, although IL-6Rα-deficient T cells mount normal primary Th1 responses in the absence of Tregs, they fail to mature into functional memory cells, demonstrating a key role for IL-6 in CD4+ T cell memory formation.

**\*For correspondence:** ruslan.medzhitov@yale.edu

†These authors contributed equally to this work

**Reviewing editor**: Tadatsugu Taniguchi, University of Tokyo, Japan

## Introduction

Innate immune recognition is mediated by a variety of pattern recognition receptors (PRRs), including Toll-like receptors (TLRs) (*Iwasaki and Medzhitov, 2004*), NOD-like receptors (NLRs) (*Martinon et al., 2009*), C-type lectin receptors (CLRs) (*Geijtenbeek and Gringhuis, 2009*), and retinoic acid inducible gene-I (RIG-I)-like receptors (RLRs) (*Pichlmair and Reis e Sousa, 2007*), all of which recognize conserved molecular structures specific to microbes (pathogen-associated molecular patterns, or PAMPs) and activate adaptive immune responses through the induction of DC maturation. The DC maturation process involves a redistribution of major histocompatibility complex (MHC) molecules from intracellular endocytic compartments to the cell surface, increased expression of costimulatory molecules, and secretion of inflammatory cytokines and chemokines, which is essential for the activation of naive T cells (*Banchereau and Steinman, 1998*). Previously, we demonstrated that TLR-induced DC maturation and migration to the draining lymph nodes, in the absence of TLR-induced inflammatory cytokines, is insufficient for the induction of T cell activation (*Pasare and Medzhitov, 2004*). T cell responses are controlled by naturally occurring CD4+ CD25+ regulatory T cells (Tregs), which play an important role in maintaining immune tolerance and homeostasis. Accordingly, the absence of Foxp3, the lineage-defining transcription factor that is critical for the generation and function of Tregs results in the development of a fatal autoimmune disease (*Fontenot and Rudensky, 2005*; *Sakaguchi, 2005*). We previously found that cytokines produced by DCs following TLR activation are critical for releasing responder CD4+ T cells from suppression by Tregs (*Pasare and Medzhitov, 2003*). Based on in vitro experiments as well as the analysis of complete IL-6 KO mice, we implicated IL-6 as a mediator for this

**eLife digest** The human body's ability to defend itself against pathogens relies on two distinct but connected systems: the innate and the adaptive immune systems. Innate immune cells survey their environment and use receptors located on their surface to distinguish between molecules that are harmless and molecules that stem from pathogens. When the cells of the innate immune system detect a pathogen, they secrete signaling molecules to alert adaptive immune cells to the invaders. Both sets of immune cells then mount a coordinated attack that usually kills the pathogen.

The adaptive immune system also produces memory cells that retain information about the pathogen: this allows the organism to mount a fast and efficient immune response the next time the same type of pathogen strikes. However, it is not completely understood how the innate immune system communicates with the adaptive immune system to allow these processes to take place.

One of the signaling molecules involved in the communication between different types of immune cells is a protein called Interleukin 6 (IL-6). This protein must be produced in order to trigger the immune response: however, many immune cells are able to recognize and respond to IL-6, so it has been difficult to study its impact on specific cell types.

Nish et al. have now investigated the effects of IL-6 on T cells, one of the main types of adaptive immune cell, by creating mice with T cells that are not able to recognize IL-6. The detection of pathogens by innate immune cells normally has several effects: the population of T cells increases; the T cells produce daughter cells—T helper cells—that support innate immune cells in killing pathogens; and memory cells are formed. Nish et al. find that these responses are impaired in the mutant mice.

To understand why, Nish et al. turn to T regulatory cells; these are adaptive immune cells that control the strength of the immune response. These experiments show that when T cells are 'blind' to IL-6, they are more sensitive to the action of T regulatory cells, and this disturbs the delicate balance between the stimulation and inhibition of the immune system. Nish et al. go on to show that IL-6 works together with another signaling molecule, Interleukin 1, to regulate how the T cells respond. The work helps to explain how the adaptive immune system mounts an immune response against pathogens but not against the host's own tissues.

block of suppressor activity (*Pasare and Medzhitov, 2003*). However, the pleiotropic nature of IL-6 has made it difficult to assess the T cell-specifc function of this cytokine in vivo.

IL-6 has been previously described as a B cell growth factor, an initiator of acute phase responses, and a mediator of T cell survival (*Kamimura et al., 2003*). Recently, IL-6 has also been implicated in the differentiation of specific $CD4^+$ T cell subsets. IL-6 has been demonstrated to be an important cytokine that governs the differentiation of Th17 cells. In particular, it is thought to regulate the switch between the induction of $Foxp3^+$ Tregs and Th17 cells (*Bettelli et al., 2006*; *Veldhoen et al., 2006*). Stimulating T cells with TGF-β results in the induction of Foxp3, whereas combining TGF-β with IL-6 represses Foxp3 expression and induces retinoid-related orphan receptor (ROR)-γt, resulting in the differentiation of Th17 cells (*Ivanov et al., 2006*). IL-6 has also been suggested to be essential for the differentiation of the T follicular helper (Tfh) cell lineage, which is defined by expression of the markers CXCR5 and programmed death receptor-1 (PD-1). Treating T cells with IL-6 leads to the upregulation of the transcriptional repressor B cell lymphoma 6 (Bcl-6), a process that is thought to drive Tfh cell generation (*Johnston et al., 2009*; *Nurieva et al., 2009*). Tfh cells specialize in providing help to germinal center (GC) B cells (*Vinuesa et al., 2005*; *King et al., 2008*).

The IL-6 receptor complex consists of the ligand-binding subunit, the IL-6Rα chain, and the signal-transducing subunit, gp130 (*Taga et al., 1989*), which is a common signaling transducer for several cytokines including IL-6, IL-11, leukemia inhibitory factor (LIF), oncostatin M (OSM), cardiotrophin-1 (CT-1), and IL-35. Homodimerization of gp130 upon IL-6 binding results in the activation of the gp130-associated tyrosine kinases JAK1, JAK2 and TYK2 and the subsequent phosphorylation of the transcription factors STAT1 and STAT3 (*Kamimura et al., 2003*). In addition to the membrane-bound receptor, a soluble form of the IL-6Rα chain can be generated through either alternative splicing or proteolytic cleavage and is capable of binding to IL-6 and activating cells through association with gp130 (*Kamimura et al., 2003*). Although both IL-6-deficient mice and mice bearing a T cell-specific

deletion of gp130 have been used to examine the in vivo function of IL-6 in T cell responses, the pleiotropic nature of IL-6, as well as the promiscuous use of the gp130 signaling subunit, has complicated these analyses (*Kamimura et al., 2003*). Thus, the T cell-specific role of IL-6 signaling in vivo has remained poorly understood.

In the present study, we therefore used mice carrying a T cell-specific ablation of the IL-6Rα chain, in order to investigate the role of IL-6 in the induction of CD4+ T cell responses following TLR activation. Our data not only confirm an important function for IL-6 in Th17 cell differentiation, but also reveal a critical role for IL-6 in the induction of the Th1 cell response in vivo. IL-6 enables T cell activation by acting on responder CD4+ T cells to make them less sensitive to the suppressive activity of Tregs, in part by blocking Treg-mediated inhibition of IL-2Rα expression in responder CD4+ T cells in cooperation with IL-1β. Although not absolutely required for the generation of Tfh cells, we also found that IL-6 signaling is important for the ability of these cells to provide help to B cells. In addition, we reveal a role for IL-6 in the generation of functional memory CD4+ T cells.

## Results

### CD4+ T cell responses are impaired in the absence of T cell-specific IL-6 signaling

To examine the function of IL-6 in CD4+ T cell responses in vivo, we generated mice in which T cells were specifically deficient of the IL-6Rα by crossing a floxed allele of *Il6ra* with mice expressing the Cre recombinase under the control of the CD4 promoter (hereafter called IL-6Rα^T-KO mice). Since the Cre-encoding transgene is expressed at the double positive stage in thymic development, both CD4+ and CD8+ T cells in the periphery of IL-6Rα^T-KO mice failed to express the IL-6Rα (*Figure 1A*). Importantly, both CD4+ and CD8+ T cells from IL-6Rα^T-KO mice remained deficient of the IL-6Rα after immunization with Ovalbumin (OVA) and LPS emulsified in Incomplete Freund's Adjuvant (IFA) as a carrier, suggesting that the release of the soluble form of the IL-6Rα during the immune response does not restore IL-6 signaling in these cells (*Figure 1A*). Furthermore, IL-6-induced STAT3 phosphorylation was blocked in IL-6Rα-deficient CD4+ and CD8+ T cells compared to control wild-type (WT) T cells (*Figure 1B*). To evaluate whether deficiency of the IL-6Rα on CD4+ T cells compromised the gp130-dependent signaling axis, we stimulated CD4+ T cells in vitro with α-CD3e and α-CD28 mAbs in the presence of gp130-dependent cytokines and measured the phosphorylation of STAT3 1 hr later by Western blot. Addition of IL-6 to the cells phosphorylated STAT3 very effectively in WT cells but not in IL-6Rα-deficient cells, thus confirming the results obtained by flow cytometry (*Figure 1—figure supplement 1*). Importantly, the addition of the soluble form of the IL-6Rα (sIL6Rα) together with IL-6 rescued the phosphorylation of STAT3 in IL-6Rα-deficient CD4+ T cells whereas IL-11, OSM, or CNTF did not phosphorylate STAT3 in either wild-type or IL-6Rα-deficient CD4+ T cells (*Figure 1—figure supplement 1*). These results suggest that the STAT3-dependent signaling pathway remains intact in IL-6Rα-deficient CD4+ T cells and that other tested cytokines of the IL-6 family do not play a major role in the activation of naive CD4+ T cells. We therefore demonstrate efficient deletion of the IL-6Rα and abrogation of IL-6 signaling in T cells from IL-6Rα^T-KO mice.

Prior studies suggested that IL-6 is a mediator of T cell survival. Specifically, IL-6 has been shown to protect CD4+ T cells from α–CD3-induced apoptosis and Fas-mediated cell death in vitro (*Takeda et al., 1998*). Moreover, complete IL-6-deficient mice were reported to have reduced T cell numbers in the thymus and peripheral lymphoid organs (*Kamimura et al., 2003*). We therefore examined how IL-6Rα deficiency affects T cell homeostasis. We verified that complete IL-6-deficient mice had decreased numbers of T cells (*Figure 1—figure supplement 2A*). In contrast, we found that the absolute numbers of CD4+ and CD8+ T cells in the thymus and lymph nodes of IL-6Rα^T-KO mice were similar to that of WT mice (*Figure 1—figure supplement 2A*). Consistent with WT levels of CD4+ and CD8+ T cells in IL-6Rα^T-KO mice, we did not observe an increased tendency of IL-6Rα-deficient CD4+ T cells to undergo apoptosis. These cells became activated (CD44^hi, CD62L^lo) and cleaved caspase-3 to the same extent as control cells upon stimulation with α-CD3 and α-CD28 in vitro, irrespective of the presence of IL-6 in the culture medium (*Figure 1—figure supplement 2C*). Likewise, CD4+ T cells from the thymus or peripheral lymphoid organs of IL-6Rα^T-KO mice stained positive for annexin V in similar proportions as WT CD4+ T cells (*Figure 1—figure supplement 2B*). Taken together, these findings demonstrate that IL-6 signaling in T cells is dispensable for T cell homeostasis. Thus, the reduced T cell numbers in complete IL-6-deficient mice are likely a consequence of IL-6 regulating T cell homeostasis indirectly through other cell types.

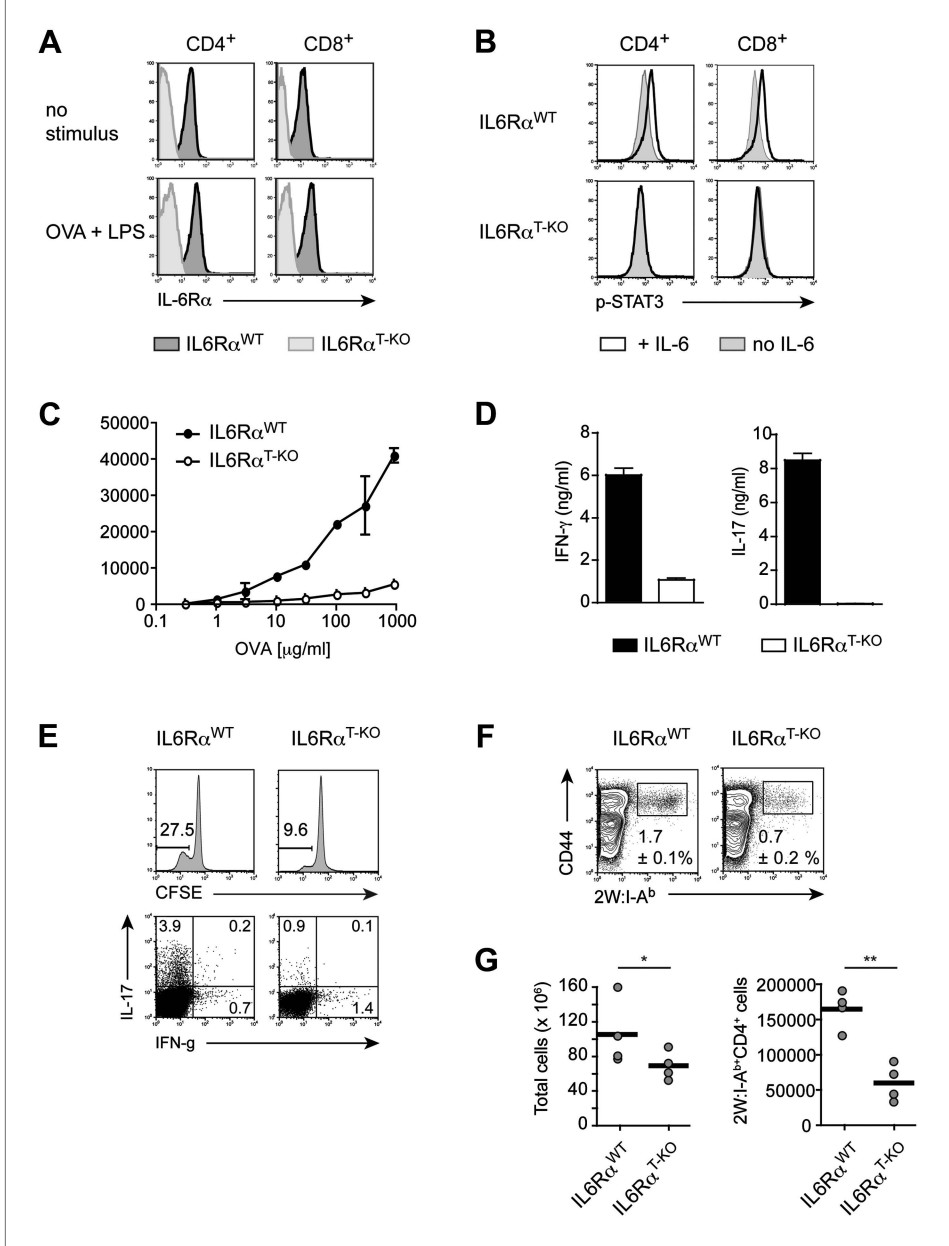

**Figure 1**. Impairment of both Th1 and Th17 responses in IL-6Rα T-KO mice. (**A**) Expression of the IL-6Rα chain by CD4+ and CD8+ T cells from WT and IL-6Rα T-KO mice was examined by flow cytometry in naive mice (upper panels) and in mice immunized with OVA plus LPS in IFA (lower panels). (**B**) CD4+ and CD8+ T cells purified from WT and IL-6Rα T-KO mice were either left untreated (shaded histogram) or stimulated with recombinant IL-6 for 20 min (open histogram) and expression of phosphorylated STAT3 (Y705) was assessed by flow cytometry. (**C**) CD4+ T cells were purified from the popliteal and inguinal lymph nodes of WT and IL-6Rα T-KO mice 7 days following immunization in the footpads with OVA and LPS emulsified in IFA. Proliferation was assessed by [3H]-thymidine incorporation following coculture of purified CD4+ T cells with irradiated splenocytes presenting titrating doses of OVA for approximately 72–84 hr. (**D**) Supernatants of CD4+ T cells from immunized mice were collected approximately 84 hr after restimulation with antigen in vitro. The production of IFN-γ and IL-17 by CD4+ T cells was examined by ELISA. (**E**) Proliferation and cytokine expression were measured by CFSE-labeling and intracellular cytokine staining, respectively, 72 hr after in vitro restimulation. Stimulations were performed as described in (**C**). (**F**) Day 7 following immunization with 2W peptide and LPS emulsified in IFA, the percentages of antigen-specific T cells were determined by 2W:I-Ab tetramer staining. Gated on total CD4+ cells. (**G**) Total cell numbers and absolute numbers of 2W:I-Ab tetramer positive CD4+ T cells in the draining lymph nodes of WT and IL-6Rα T-KO mice after the immunization. *Figure 1. Continued on next page*

*Figure 1. Continued*

Data are representative of three independent experiments. Line graphs and bar graphs represent mean ± SEM; for all panels: p≤0.05.

The following figure supplements are available for figure 1:

**Figure supplement 1**. Phosphorylation of STAT3 in IL6Rα-deficient CD4+ T cells after stimulation through gp130-containing receptors.

**Figure supplement 2**. T cell homeostasis is unaffected when IL-6 signaling is abrogated specifically in T cells.

**Figure supplement 3**. IL-6Rα-deficient CD4+ CD45RB^hi T cells fail to induce Colitis and IL-6Rα^T-KO mice are resistant to EAE.

**Figure supplement 4**. IL-6Rα^T-KO mice show impaired CD4+ T cell responses.

**Figure supplement 5**. Defective CD4+ T cell response in IL-6Rα^T-KO mice immunized with 2W peptide plus LPS in IFA.

**Figure supplement 6**. BrdU and active Caspase-3 staining in the presence or absence of T cell-intrinsic IL-6 signaling.

Next, we determined whether IL-6 signaling in T cells is required for the initiation of CD4+ T cell responses by immunizing IL-6Rα^T-KO and WT mice with OVA plus LPS in IFA. 7 days following immunization, CD4+ T cells purified from the popliteal and inguinal lymph nodes were co-cultured with irradiated WT splenocytes from naïve mice, as antigen-presenting cells, in order to examine the recall response to OVA. We found that T cells isolated from IL-6Rα^T-KO mice showed impaired proliferation upon restimulation with the antigen as measured by [$^3$H]-thymidine incorporation (***Figure 1C***, ***Figure 2—figure supplement 2A***). Furthermore, in contrast to WT CD4+ T cells, CD4+ T cells from IL-6Rα^T-KO mice failed to produce IL-17 (***Figure 1D***, ***Figure 2—figure supplement 2B***). In order to confirm this phenotype and characterize the IL-6Rα^T-KO mice further, we also tested for the induction of Th17-dependent autoimmune diseases in these mice. IL-6Rα-deficient CD4+ CD45RB^hi T cells failed to induce colitis when transferred into RAG2 KO mice (***Figure 1—figure supplement 3A–C***) and compared to WT mice, IL-6Rα^T-KO mice were also resistant to the induction of EAE following immunization with MOG$_{35-55}$/CFA (***Figure 1—figure supplement 3D***). Thus, the Th17 response of IL-6Rα^T-KO mice was defective, which was expected given the role of IL-6 in Th17 differentiation. However, immunization of the IL-6Rα^T-KO mice with OVA and LPS in IFA also revealed a profound defect in the Th1 response as IL-6Rα-deficient CD4+ T cells failed to secrete robust amounts of IFN-γ after restimulation with antigen (***Figure 1D***). We obtained similar results following immunization with OVA and other TLR ligands, such as peptidoglycan (PGN), CpG DNA, or the more complex Complete Freund's Adjuvant (CFA) (***Figure 1—figure supplement 4A–C***).

To address whether the defective cytokine production in in vitro restimulation assays were a consequence of defective proliferation and/or T cell differentiation, we also analyzed CD4+ T cell proliferation and cytokine production using CFSE labeling and intracellular cytokine staining, respectively. We confirmed that T cells from IL-6Rα^T-KO mice were impaired in their ability to proliferate following restimulation (***Figure 1E***). Although we observed an overall reduction in T cell proliferation, a small fraction of T cells from IL-6Rα^T-KO mice proliferated normally as assessed by CFSE dilution (***Figure 1E***) and were able to produce IFN-γ at similar levels as WT cells. The cells remained deficient in IL-17, regardless of proliferation (***Figure 1E***). These results suggest that the defect in the IFN-γ response is not due to a requirement for IL-6 in the differentiation of IFN-γ-producing cells. Instead, our results imply that the IFN-γ response is deficient in IL-6Rα^T-KO mice as a consequence of impaired T cell proliferation. Conversely, the deficient IL-17 response, regardless of proliferative status, reflects a dual requirement of IL-6 for both CD4+ T cell proliferation and differentiation of IL-17 producing T cells.

The experiments so far assessed the CD4+ T cell response after restimulation with OVA in vitro. Staining CD4+ T cells with MHC class II tetramers is a way to track antigen-specific CD4+ T cells more directly. In contrast to class II tetramers using OVA-derived peptides, 2W:I-A^b tetramers, which stain CD4+ T cells that are specific for the 2W peptide, offer a better staining performance. Furthermore,

naive 2W-specific CD4$^+$ T cells are more frequent than OVA-specific CD4$^+$ T cells, which facilitates the in vivo analysis of the CD4$^+$ T cell response (*Moon et al., 2007*). We therefore opted to track antigen-specific T cells with 2W:I-A$^b$ tetramers following immunization with 2W peptide and LPS in IFA (*Moon et al., 2007*). To ensure that the choice of a different antigen did not affect the phenotype of IL-6Rα$^{T-KO}$ mice, we first immunized the mice with the 2W peptide and measured the proliferation and IFN-γ secretion following restimulation in vitro. IL-6Rα-deficient CD4$^+$ T cells did not expand and secrete IFN-γ efficiently (*Figure 1—figure supplement 5A,B*). Antigen-specific 2W:I-A$^{b+}$CD4$^+$ T cells from IL-6Rα$^{T-KO}$ mice were also less frequent in these cultures compared to WT controls (*Figure 1—figure supplement 5C*). These results resembled those obtained with OVA-based immunization and demonstrated that the choice of antigen did not affect the outcome of the CD4$^+$ T cell response. We thus tracked antigen-specific 2W:I-A$^{b+}$ CD4$^+$ T cells directly in vivo. We found that the percentage of 2W:I-A$^{b+}$ CD44$^+$ CD4$^+$ T cells was reduced in IL-6Rα$^{T-KO}$ mice compared with the WT controls (*Figure 1F*). Moreover, relative to WT mice, IL-6Rα$^{T-KO}$ mice had fewer numbers of cells in their lymph nodes on day 7 after immunization (*Figure 1G*), which resulted in lower absolute numbers of antigen-specific T cells in IL-6Rα$^{T-KO}$ mice (*Figure 1G*). Although the percentage and absolute number of 2W:I-A$^{b+}$CD4$^+$ T cells were significantly reduced in IL-6Rα$^{T-KO}$ mice on day 7 following immunization, we found that the small proportion of remaining IL-6Rα-deficient CD4+ T cells in these mice proliferated to a similar extent as the WT, if not slightly better, as assessed by BrdU incorporation (*Figure 1—figure supplement 6A*). Consistently, we failed to see a difference in cell death between WT and IL-6Rα-deficient CD4+ T cells on day 7 following immunization as determined by active Caspase-3 staining (*Figure 1—figure supplement 6B*). Collectively, these results confirm the expected role of IL-6 signaling in Th17 differentiation, but they also reveal a role for IL-6 signaling in regulating T cell expansion and the subsequent induction of Th1 responses in vivo.

## IL-6 signaling is required in CD4$^+$ T cells to render them refractory to Treg-mediated suppression in vivo

We previously showed that IL-6 produced by DCs following TLR activation is required to overcome Treg-mediated suppression in vitro (*Pasare and Medzhitov, 2003*). Therefore, we asked whether impaired T cell responses in IL-6Rα$^{T-KO}$ mice are due to unopposed suppression by Tregs in the absence of IL-6 signaling. To test this possibility, IL-6Rα$^{T-KO}$ mice were either left untreated or injected with a monoclonal antibody (mAb) against CD25, which resulted in the depletion of approximately 80% of Foxp3$^+$ Tregs (*Figure 2—figure supplement 1A,B*) prior to immunization with OVA and LPS in IFA. While CD4$^+$ T cells from IL-6Rα$^{T-KO}$ mice with an intact Treg compartment showed impaired proliferation and production of IFN-γ upon restimulation, these responses were largely rescued upon depletion of Tregs (*Figure 2A,B*, *Figure 2—figure supplement 2A,B*). This result suggests that IL-6 may control T cell responses by overcoming the suppressive effect of Tregs. Although the IFN-γ response was restored in the absence of Tregs, the IL-17 response remained defective in these mice, confirming the nonredundant role of IL-6 in Th17 differentiation in vivo, at least under the experimental conditions used in this study.

Because both conventional CD4$^+$ T cells and Tregs are unresponsive to IL-6 in IL-6Rα$^{T-KO}$ mice, we wanted to determine which T cell subset was targeted by IL-6 to mediate its effects. One possibility is that IL-6 targets conventional CD4$^+$ T cells in order to make them refractory to Treg-mediated suppression. Therefore, in the absence of IL-6 signaling, CD4$^+$ T cells would remain susceptible to suppression by Tregs. Alternatively, IL-6 might target Tregs directly to block their suppressive activity. To test this latter possibility, we generated IL-6Rα$^{FL/FL}$; Foxp3$^{Cre/+}$ mice (hereafter called IL-6Rα$^{TREG-KO}$ mice) to specifically delete the IL-6Rα in Tregs. We found that CD4$^+$ T cells from IL-6Rα$^{TREG-KO}$ mice proliferated normally and produced both IFN-γ and IL-17 at similar levels as WT CD4$^+$ T cells following immunization with OVA and LPS in IFA (*Figure 2C,D*). These findings demonstrate that, at least under the experimental conditions used here, IL-6 signaling is not required in Tregs, and instead IL-6 acts on conventional CD4$^+$ T cells to make them less sensitive to the suppressive effect of Tregs.

To further rule out the possibility that the difference that we see in the response of CD4$^+$ T cells in IL-6Rα$^{T-KO}$ mice after immunization is due to altered Treg function in the absence of IL-6 signaling, we sorted CD4$^+$CD25$^+$ IL-6Rα-deficient Tregs to examine their ability to suppress proliferation of purified conventional C4$^+$CD25$^-$ T cells. We found that IL-6Rα-deficient Tregs were able to inhibit the proliferative response of WT responder CD4$^+$ T cells to the same degree as WT Tregs (*Figure 2—figure supplement 3*).

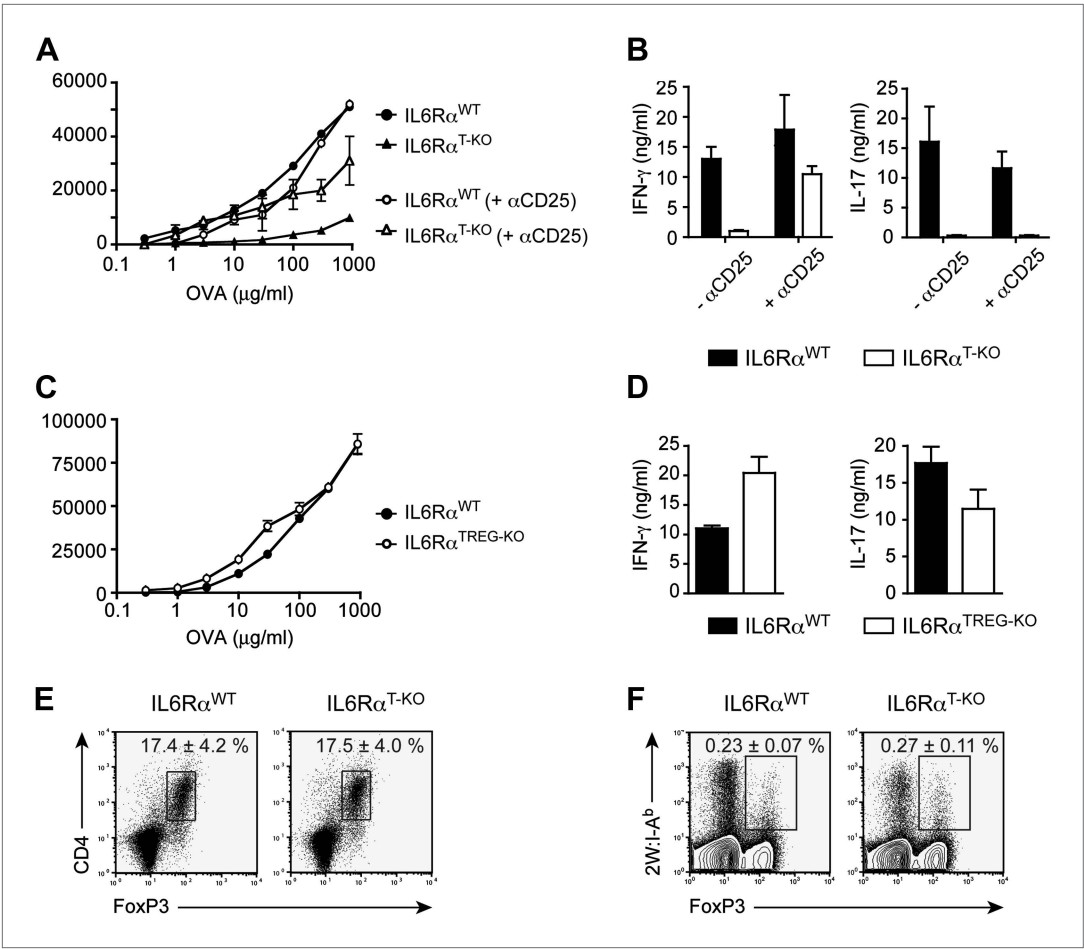

**Figure 2**. IL-6 signaling in responder T cells is required to overcome suppression by Tregs. (**A** and **B**) WT and IL-6RαT-KO mice received a single intravenous injection of α-CD25 monoclonal antibody 3 days prior to immunization to transiently deplete Tregs. Mice were immunized in the footpads with OVA and LPS emulsified in IFA and 7 days following immunization, purified CD4+ T cells were restimulated as in **Figure 1C**. Proliferation (**A**) and cytokine production (**B**) were measured by [3H]-thymidine incorporation and ELISA, respectively, 72–84 hr following restimulation. (**C** and **D**) WT and IL-6RαTREG-KO mice were immunized with OVA and LPS emulsified in IFA and purified CD4+ T cells were restimulated 7 days following immunization as in **Figure 1C**. Proliferation (**C**) and production of IFN-γ and IL-17 (**D**) were measured as before. (**E**) Percentages of Foxp3+ CD4+ T cells in the draining lymph nodes of WT and IL-6RαT-KO mice 7 days following immunization with OVA and LPS in IFA. (**F**) WT and IL-6RαT-KO mice were immunized with 2W peptide and LPS emulsified in IFA and 7 days after immunization cells isolated from the draining lymph nodes were stained with 2W:I-Ab tetramer to determine percentages of CD4+ Foxp3+ 2W:I-Ab+ cells. Gated on total CD4+ cells. Data are representative of at least three independent experiments. Line graphs and bar graphs represent mean ± STD.

The following figure supplements are available for figure 2:

**Figure supplement 1**. Treg depletion efficiency with α-CD25 treatment.

**Figure supplement 2**. Statistical representation of the CD4+ T cell response in the presence or absence of Tregs.

**Figure supplement 3**. Suppression of T cell proliferation by WT and IL-6RαT-KO regulatory cells.

**Figure supplement 4**. Absolute cell numbers of antigen-specific Foxp3+ Tregs in IL-6RαT-KO and control mice.

Several studies have shown reciprocal regulation of Tregs and Th17 cells and demonstrated that IL-6 inhibits TGF-β-driven induction of Tregs and induces the production of IL-17 (**Bettelli et al., 2006**; **Veldhoen et al., 2006**). Moreover, mice bearing a T cell-specific deletion of gp130 as well as

IL-6-deficient mice have been shown to have elevated numbers of Foxp3$^+$ Tregs in their secondary lymphoid organs, although this has not been universally observed for the latter mouse strain (*Korn et al., 2007*, *2008*). Therefore, it remained possible that in the absence of the IL-6Rα, Tregs were abnormally expanded in vivo, resulting in the defective T cell proliferation observed in *Figure 1*. To address this, we immunized IL-6Rα$^{T-KO}$ mice with OVA and LPS in IFA and analyzed the percentages of Foxp3$^+$ T cells. We found no significant increase in the proportion of these cells among CD4$^+$ T cells, compared to the WT (*Figure 2E*). In support of this notion, we also did not detect significant expansion of antigen-specific Foxp3$^+$ T cells following immunization with 2W peptide and LPS in IFA and the absolute numbers of Tregs were lower in IL-6Rα$^{T-KO}$ mice compared with WT controls (*Figure 2F*, *Figure 2—figure supplement 4*). Altogether, these results suggest that expansion of induced Tregs or impairment of their function in the absence of IL-6 signaling does not account for the diminished CD4$^+$ T cell response, at least under the experimental conditions used here. Instead, IL-6 signaling is required in CD4$^+$ T cells to render them refractory to Treg-mediated suppression.

## IL-6 signaling is not essential for Tfh cell development, but is important for their function

Induction of CD4$^+$ T cell responses is essential for mounting productive T cell-dependent B cell responses. Since Tfh cells play a critical role in providing help for GC B cells, we wished to characterize the role of IL-6 in regulating this aspect of CD4$^+$ T cell biology. It was previously shown that relative to WT controls, complete IL-6-deficient mice have a strong decrease in the frequency of Tfh cells following protein immunization (*Nurieva et al., 2008*). However, when we immunized IL-6Rα$^{T-KO}$ mice with OVA and LPS in IFA and examined Tfh cell percentages 7 days later by staining for CXCR5 and PD-1, we found only a modest reduction of Tfh cells in IL-6Rα$^{T-KO}$ mice. In these mice, the frequency of Tfh cells, which were also ICOS$^{hi}$ and PSGL-1$^{low}$, was reduced by approximately 25% compared to WT controls (*Figure 3A,B*), suggesting that IL-6 signaling in T cells does not play an essential role in Tfh cell development, and that IL-6 may affect Tfh cell development indirectly. We then examined Tfh cell function by several approaches. First, we determined the structure of the GCs and the location of CD4$^+$ T cells within the GCs in IL-6Rα$^{T-KO}$ and WT controls by immunofluorescence. However, we did not observe obvious changes in the GC structure in IL-6Rα$^{T-KO}$ mice 14 days after immunization with OVA plus LPS in IFA (*Figure 3C*). We also did not observe a strong reduction in the number of CD4$^+$ T cells within the GCs of IL-6Rα$^{T-KO}$ mice (*Figure 3D*). Next, we analyzed the expression of Bcl-6, the lineage-defining transcription factor for Tfh cells, and IL-21, the Tfh cell-associated cytokine essential for GC formation and antibody production. Indeed, Tfh cells from both WT and IL-6Rα$^{T-KO}$ mice upregulated both Bcl6 and IL-21 when compared to non-TFH cells (*Figure 3E,F*). However, IL-6Rα-deficient Tfh cells failed to reach WT levels of expression for either of the two genes, suggesting a defect in the function of these cells in IL-6Rα$^{T-KO}$ mice.

To directly test the possibility that reduced expression of Bcl6 and IL-21 in IL-6Rα-deficient Tfh cells has functional consequences for GC formation and the resulting antibody response, we tested these responses in IL-6Rα$^{T-KO}$ mice after immunization with OVA and LPS in IFA. Interestingly, although significant numbers of Tfh cells were generated in the absence of T cell-specific IL-6 signaling, the fraction of PNA$^+$Fas$^+$CD19$^+$ GC cells was reduced by approximately 50% (*Figure 4A,B*). Moreover, IL-6Rα$^{T-KO}$ mice also exhibited decreased percentages of CD138$^+$B220$^-$ plasma cells following immunization (*Figure 4C,D*). We then examined the antigen-specific antibody production by ELISA and detected a moderate reduction in the amount of antigen-specific IgG1 in the absence of IL-6 signaling in T cells (*Figure 4E*). However, the titers of OVA-specific IgG2c, the isotype regulated by IFN-γ, were significantly reduced in these mice compared to WT controls (*Figure 4E*).

Since our results suggested that IL-6 is required for T cells to overcome Treg-mediated suppression, we asked whether the effects of deficient IL-6 signaling in CD4$^+$ T cells on the B cell response are conditional on the presence of Tregs. To test this possibility, we depleted IL-6Rα$^{T-KO}$ and control mice of Tregs prior to immunization with OVA and LPS in IFA and examined GC formation and antigen-specific antibody production. Treg depletion indeed restored the size of the GC compartment and OVA-specific IgG2c responses in IL-6Rα$^{T-KO}$ mice (*Figure 4F*, *Figure 4—figure supplement 1*). Collectively, these results suggest that while not essential for Tfh cell development, IL-6 signaling in T cells is important for their ability to provide help to B cells, presumably in part by overcoming suppression by Tregs.

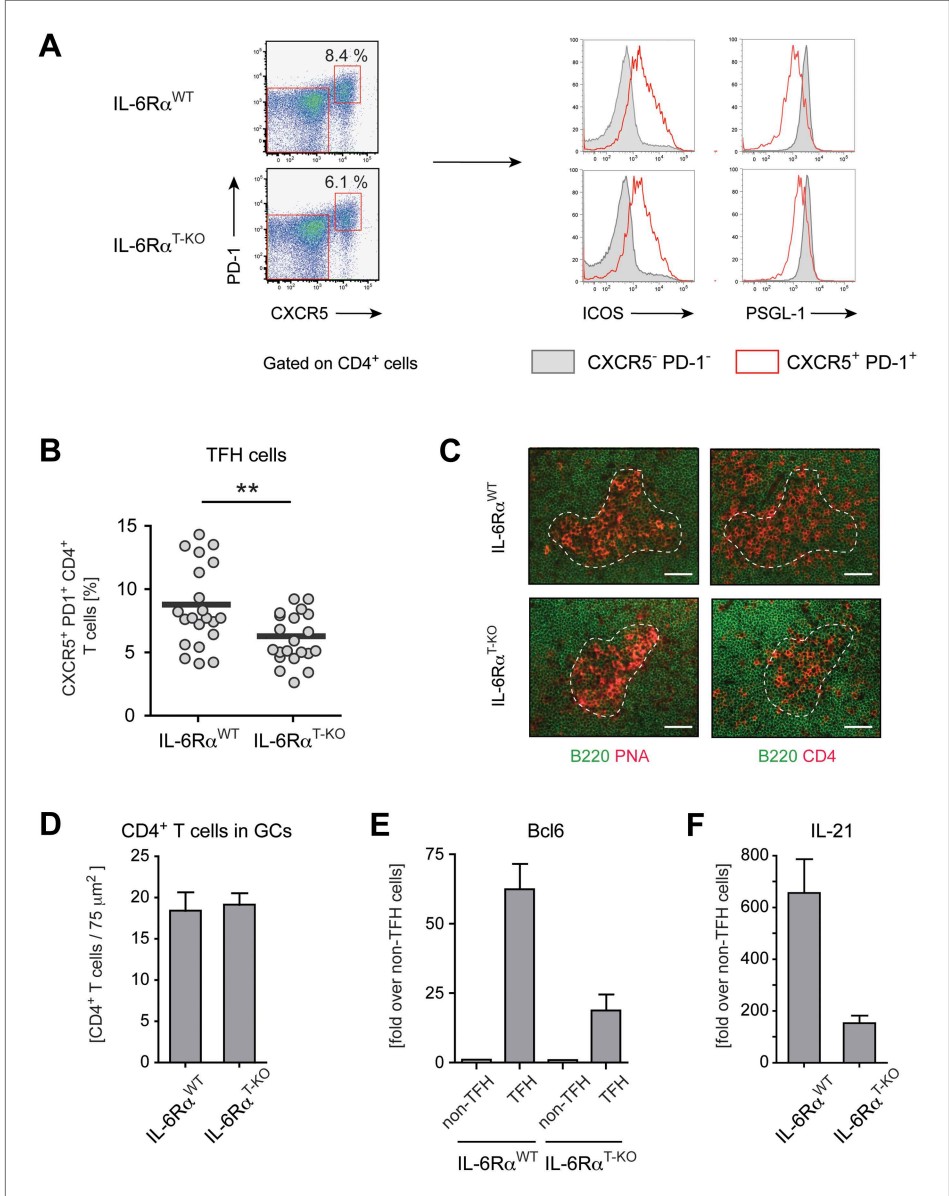

**Figure 3**. Tfh cells generated in the absence of IL-6 signaling have reduced expression of Bcl-6 and IL-21.
(**A**) Frequency of CXCR5hi PD-1hi CD4+ Tfh cells in draining lymph nodes of WT and IL-6RαT-KO mice 7 days following immunization with OVA and LPS in IFA (left panels). Additional surface markers expressed by Tfh cells from IL-6RαT-KO and control mice (right panels). (**B**) Statistical representation of the frequency of Tfh cells in IL-6RαT-KO and control mice. Shown are the combined results of multiple experiments, each closed circle represents one mouse, p≤0.005. (**C**) GC structure in the lymph nodes of IL-6RαT-KO and control mice of immunized mice. Adjacent tissue sections were stained for PNA (red) and B220 (green), or CD4 (red) and B220 (green). The dashed line demarcates the approximate border of the GC and was used to identify the GC location in the stainings with mAbs again B220 and CD4 (in addition to a marked decrease of B220 signal at the site of the GCs). Representative images are shown. (**D**) Number of CD4+ T cells within the GCs per 75 μm². (**E** and **F**) Quantitative PCR measuring the expression of Bcl-6 (**E**) and IL-21 (**F**) in sorted CXCR5hi PD-1hi CD4+ Tfh and CXCR5low PD-1low CD4+ non-Tfh cells from immunized IL-6RαT-KO and control mice. Data show fold-induction over non-Tfh cells. A representative out of three independent experiments is shown.

## IL-6 signaling in CD4+ T cells is required for the generation of functional memory CD4+ T cells

The mechanism by which CD4+ T cells differentiate into long-lived memory cells remains poorly understood (*Kaech et al., 2002*). Previously, we demonstrated that while depletion of CD4+ CD25+ Tregs in

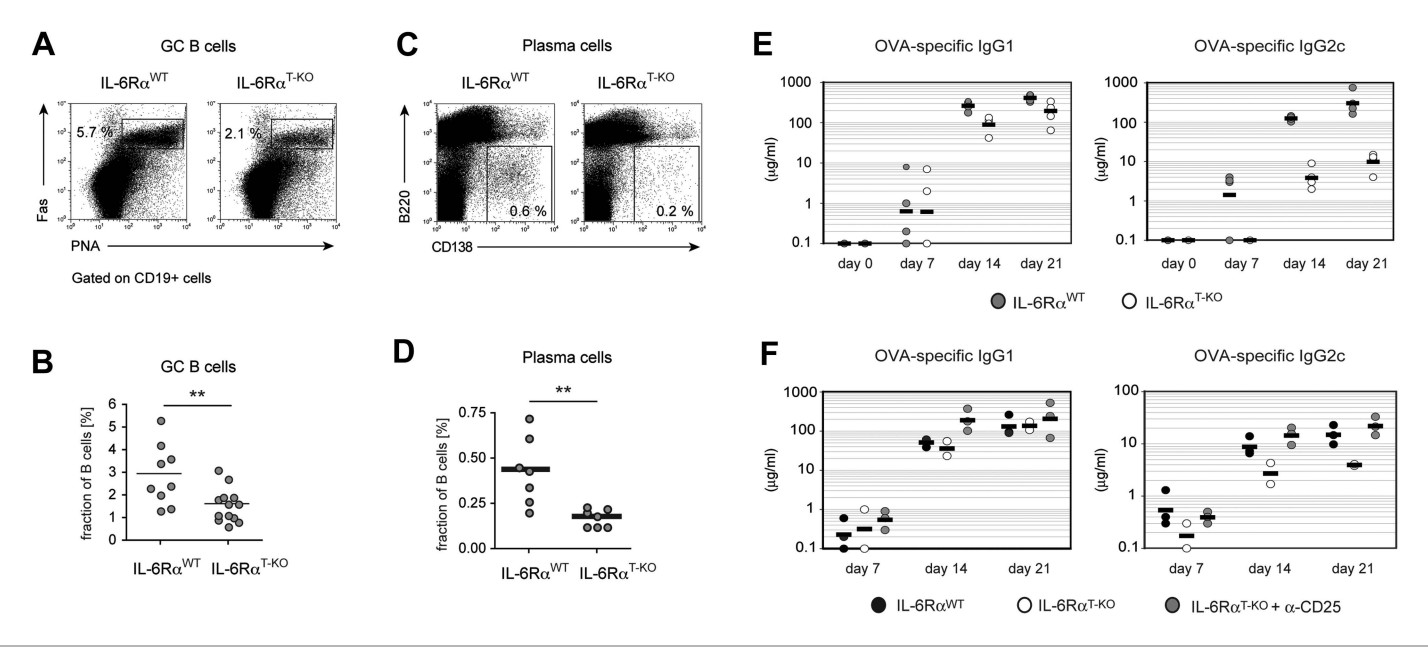

**Figure 4**. B cell responses are impaired in IL-6Rα[T-KO] mice. (**A**) Percentage of PNA[+] Fas[+] CD19[+] GC B cells in the draining lymph nodes of WT and IL-6Rα[T-KO] mice 7 days after immunization with OVA and LPS in IFA. (**B**) Statistical representation of the results shown in (**A**). Shown are the combined results of multiple experiments, each closed circle represents one mouse, p≤0.005. (**C**) Frequencies of CD138[hi]B220[neg] plasma cells in the draining lymph nodes of WT and IL-6Rα[T-KO] mice on day 7 following immunization with OVA and LPS in IFA. (**D**) Statistical representation of the results shown in (**C**). Shown are the combined results of multiple experiments, each closed circle represents one mouse, p≤0.005. (**E**) Antibody response in WT and IL-6Rα[T-KO] mice in the presence of an intact Treg compartment (**E**) or following the transient depletion of Tregs (**F**). Mice were immunized with OVA and LPS in IFA and the antigen-specific antibody titers in the serum were measured by ELISA. Tregs were depleted 3 days prior to the immunization. Representative experiments of at least three are shown.

The following figure supplements are available for figure 4:

**Figure supplement 1**. Size of the GC compartment in IL-6Rα[T-KO] and control mice in the presence or absence of Tregs.

MyD88-deficient mice restored the primary Th1 cell response, the memory response remained defective in these mice after OVA and LPS in IFA immunization (*Pasare and Medzhitov, 2004*). This result suggested that a MyD88-dependent signal(s) was required for the generation of memory CD4[+] T cell responses. Because IL-6 is produced in a MyD88-dependent manner following TLR activation, we investigated whether memory CD4[+] T cell responses were dependent on IL-6. First, we determined whether IL-6 signaling is required for the generation or maintenance of memory CD4[+] T cells. We therefore immunized IL-6Rα[T-KO] mice and WT controls with 2W peptide and LPS in IFA in the presence or absence of Tregs and tracked antigen-specific T cells 30–60 days later by 2W:I-A[b] tetramer staining. IL-6Rα[T-KO] mice generated significant amounts of 2W:I-A[b]-specific memory CD4[+] T cells that in some experiments reached WT levels, regardless of whether or not Tregs were depleted (*Figure 5A*). Memory CD4[+] T cells can be divided into different subsets based on expression of CXCR5 and PD-1, which are used to identify CXCR5[−] PD-1[−] Th1 effector memory cells, CXCR5[int] PD-1[−] Th1 central memory cells and CXCR5[hi] PD-1[+] Tfh cells (*Pepper et al., 2011*). Therefore, we also stained for different populations of memory CD4[+] T cells using these markers. We found no major differences in the percentages of these subsets, when we compared IL-6Rα[T-KO] with WT mice (*Figure 5B*). Thus, the memory CD4[+] T cell compartment is largely intact in the absence of IL-6 signaling in T cells, regardless of the presence of Tregs.

To examine whether memory CD4[+] T cells generated in the absence of the IL-6Rα were functionally intact, we employed OVA or 2W-based immunization, which we used interchangeably since they behave similarly in our assays (*Figure 1—figure supplement 5*). Thus, we depleted Tregs twice, prior to the first immunization with either OVA or 2W and again prior to secondary challenge with OVA or 2W 60 days later. This was done because Tregs recover approximately 2 weeks after α-CD25 treatment.

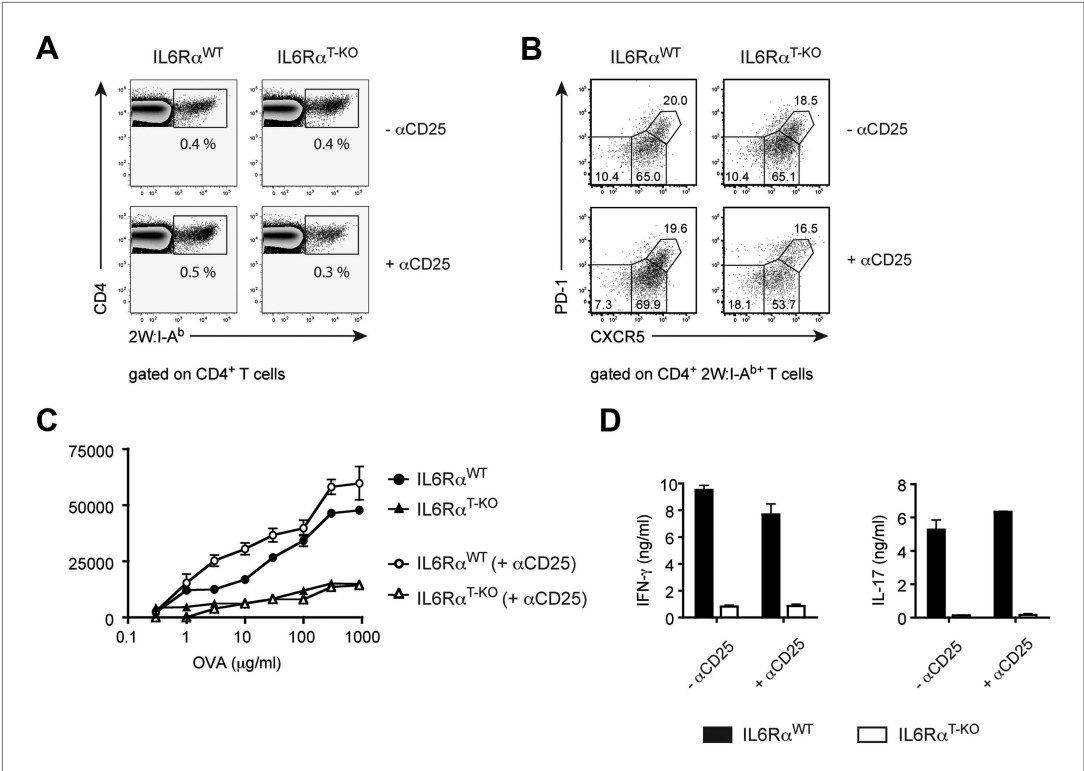

**Figure 5**. IL-6 signaling in T cells is dispensable for phenotypic, but required for functional differentiation of memory CD4+ T cells. (**A** and **B**) Generation of memory CD4+ T cells. WT and IL-6RαT-KO mice, with or without Treg depletion, were immunized once with 2W peptide and LPS in IFA. The frequency of all antigen-specific 2W:I-Ab+ CD4+ memory T cells (**A**) or individual subsets of 2W:I-Ab+ CD4+ memory T cells (**B**) in the draining lymph nodes was measured 60 days later by flow cytometry. 2W:I-Ab+ CXCR5−PD-1− represent Th1 effector memory cells, CXCR5intPD-1− represent Th1 central memory cells, and CXCR5hi PD-1+ represent Tfh cells. Data are representative of at least three independent experiments. (**C** and **D**) Expansion and cytokine secretion of memory CD4+ T cells. WT and IL-6RαT-KO mice, with or without Treg depletion, were immunized with OVA and LPS in IFA. 60 days later, mice that were previously depleted of Tregs were depleted for a second time prior to re-immunization with OVA and LPS in IFA. 7 days after the second immunization, purified CD4+ T cells were restimulated and proliferation (**C**) and cytokine production (**D**) were measured by [3H]-thymidine incorporation and ELISA, respectively. Line graph and bar graph represent mean ± SEM.

The following figure supplements are available for figure 5:

**Figure supplement 1**. Statistical representation of the CD4+ T cell memory response following immunization with OVA + LPS in IFA.

**Figure supplement 2**. Frequency of antigen-specific 2W:I-Ab+ CD4+ memory T cells in IL-6RαT-KO and control mice after the secondary immune response in the presence or absence of Tregs.

We then monitored the memory CD4+ T cell response after in vitro restimulation. Although depletion of Tregs in IL-6RαT-KO mice rescued primary CD4+ T cell proliferation and the induction of IFN-γ-producing cells following OVA and LPS in IFA immunization (**Figure 2A,B**), this was not sufficient for the generation of memory Th1 responses. Proliferation of antigen-specific CD4+ T cells and induction of the IFN-γ response was still defective in IL-6RαT-KO mice compared with their WT counterparts following secondary immunization (**Figure 5C,D**, **Figure 5—figure supplement 1**). Consistent with this finding, we also noticed a trend towards reduced frequencies of antigen-specific 2W:I-Ab+CD4+ T cells in mice 7 days after the second immunization with 2W peptide, even under conditions where Tregs were transiently absent (**Figure 5—figure supplement 2**). Collectively, these findings demonstrate that induction of a robust primary CD4+ T cell response is insufficient for the formation of the memory response and suggests that although IL-6 signaling is not critical for the generation or maintenance of memory CD4+ T cells, it is essential for their function.

# IL-6 cooperates with IL-1β to counteract Treg-mediated inhibition of IL-2R expression and signaling in responding CD4+ T cells

Our results demonstrate that IL-6 is required to overcome the suppressive effect of Tregs on T cell proliferation in vivo. To gain insights into the mechanism by which IL-6 performs this function, we cultured CD4+ T cells with Foxp3+ Tregs at a 1:1 ratio and stimulated these cells with α−CD3ε/CD28 mAbs in the presence or absence of IL-6. Since T cell responsiveness to IL-2 is important for T cell proliferation, we examined the expression of the IL-2 receptor α chain (CD25) by CD4+ Foxp3− responder T cells in the culture on day 3. We found that expression of CD25 by these cells was inhibited in the presence of Foxp3+ CD4+ T cells (*Figure 6A*). However, addition of exogenous IL-6 to these

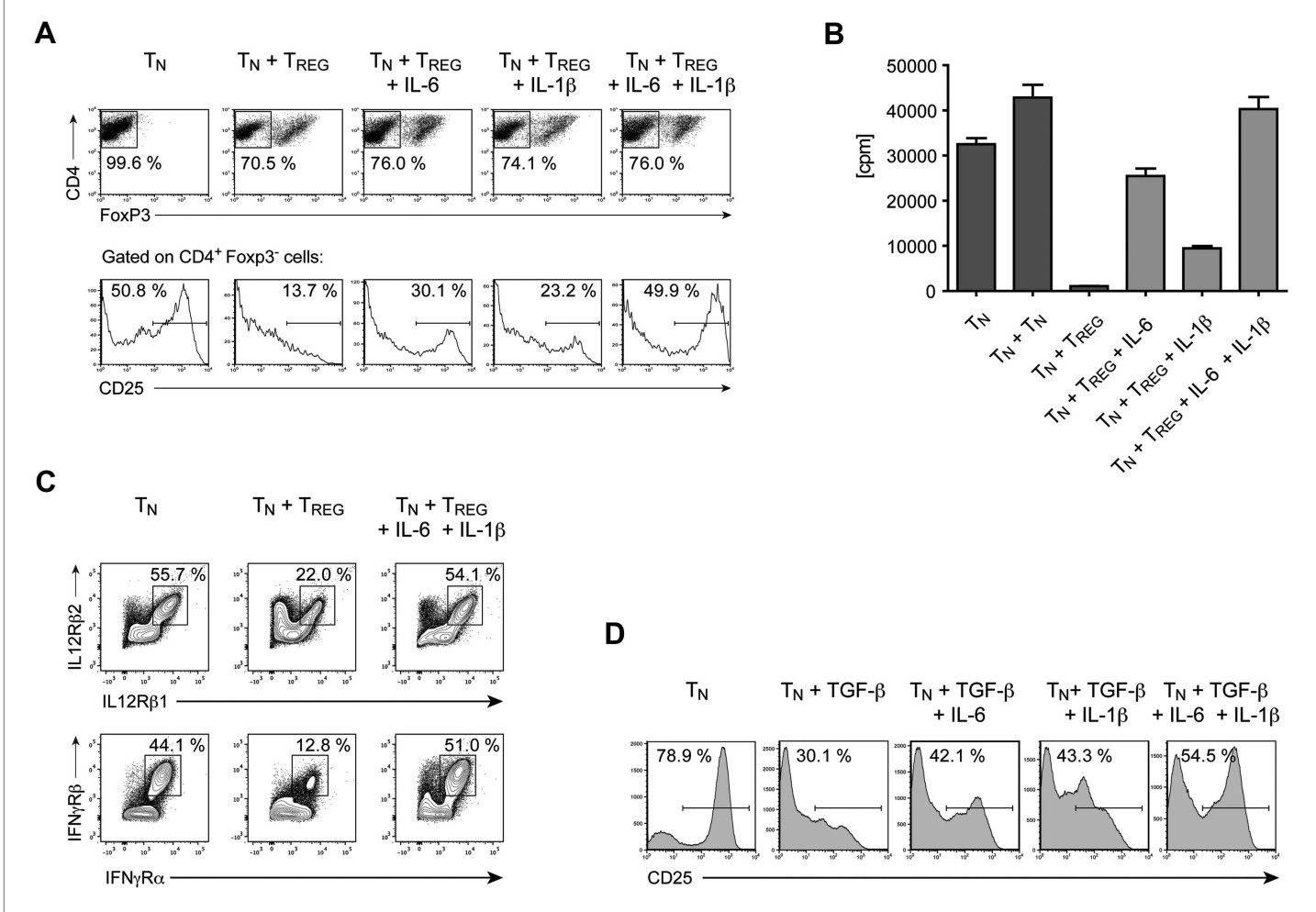

**Figure 6**. IL-6, together with IL-1β, overcomes Treg-mediated suppression by maintaining T cell responsiveness to IL-2 and other cytokines. (**A**) Purified CD4+ Foxp3- responder T cells (T_N) were either cultured alone or at a 1:1 ratio with CD4+ Foxp3+ cells and α-CD3 and α-CD28, with or without the indicated cytokines. CD25 expression by CD4+ Foxp3− cells was examined on day 3 by flow cytometry. (**B**) Proliferation of responder T cells in the presence of Tregs and indicated cytokines following stimulation with soluble α-CD3 and α-CD28. Data are representative of at least three independent experiments. (**C**) Purified CD4+ Foxp3− responder T cells (T_N) were either cultured alone or at a 1:1 ratio with CD4+ Foxp3+ cells and stimulated as in (**A**). The expression of the indicated cytokine receptors was measured by flow cytometry. (**D**) Purified CD4+ T cells were stimulated with α-CD3 and α-CD28 in the presence or absence of the indicated cytokines. Expression of CD25 by CD4+ T cells was assessed on day 4 of culture.

The following figure supplements are available for figure 6:

**Figure supplement 1**. IL-15 overcomes Treg-mediated inhibition of CD25 in responder CD4+ T cells.

**Figure supplement 2**. Pathway analysis of the differentially expressed genes in antigen-specific CD4+ T cells of IL-6Rα^T-KO mice.

cultures resulted in partial recovery of CD25 expression (*Figure 6A*). Because the effect of IL-6 was partial, we investigated whether other cytokines could fulfill a similar function as IL-6. Separate work in our laboratory indicated that IL-1 may also operate in this process (*Schenten et al., 2014*). We therefore wondered whether IL-6 and IL-1 cooperate in mediating this function. Indeed, IL-1β also partially rescued expression of CD25 by responder CD4+ T cells, although to a lesser extent than IL-6, and the combined addition of both IL-6 and IL-1β resulted in greater recovery of CD25 expression even in the presence of Tregs (*Figure 6A*). Since IL-15 is another TLR-induced cytokine that had been previously implicated in overcoming Treg-mediated suppression (*Shevach, 2009*), we also tested whether IL-15 had a similar effect on CD25 expression of CD4+ responder T cells in the presence of Tregs. Indeed, addition of IL-15 to the culture medium also resulted in the maintenance of CD25 expression (*Figure 6—figure supplement 1*).

The rescue of CD25 expression by IL-6 and/or IL-1β was accompanied by a recovery of T cell proliferation in the presence of Tregs. As expected, CD4+ responder T cells failed to proliferate following stimulation with α-CD3ε/CD28 mAbs, when equal numbers of Tregs were added to the culture (*Figure 6B*). However, the addition of either IL-6 or IL-1β resulted in a partial recovery of proliferation and the presence of both IL-6 and IL-1β in the culture medium allowed CD4+ T cells to proliferate at similar levels as CD4+ responder T cells stimulated in the absence of Tregs. These data therefore implied that the IL-6 and IL-1-induced maintenance of CD25 expression in the presence of Tregs enabled CD4+ responder T cells to proliferate.

Next, we asked whether Tregs also have an affect on the expression levels of other cytokine receptors of CD4+ responder T cells and, if so, whether IL-6 and IL-1 can reverse this effect. Indeed, Tregs suppressed the expression of the receptors for IL-12 and IFN-γ, two cytokine receptors that are essential for the generation of Th1 responses, and the suppression of these two cytokine receptors was rescued by the addition of IL-6 and IL-1 to the culture medium (*Figure 6C*). We obtained similar data for the expression of the IL-21 receptor and the common γ-chain, although the expression levels for these receptors changed more moderately (data not shown). The regulation of all these receptors, including CD25, occurred at the transcriptional level as the RNA expression of the corresponding genes largely mirrored the protein expression on the cellular surface (data not shown).

Tregs suppress T cell responses by multiple mechanisms, including via the release of inhibitory cytokines such as TGF-β and IL-10 (*Shevach, 2009*). To determine whether inhibitory cytokines associated with Tregs can suppress expression of CD25 and whether their activity is countered by IL-6, purified CD4+ T cells were stimulated in the presence of either TGF-β alone or TGF-β with IL-6. We found that treating T cells with TGF-β also resulted in impaired expression of CD25 in responder T cells (*Figure 6D*). However, the negative effect of TGF-β on CD25 expression was hindered when exogenous IL-6 was added to the culture. Additionally, we found that IL-1β also functioned similarly to IL-6 in countering the negative effect of TGF-β on expression of CD25. Moreover, adding both IL-6 and IL-1β resulted in greater recovery of CD25. Collectively, our results suggest that IL-6 cooperates with IL-1β to counter Treg suppression by maintaining T cell responsiveness to IL-2 and other cytokines important for the proliferation and differentiation of CD4+ T cells. This mode of action for IL-6 and IL-1β may therefore indeed be a part of the mechanism that renders CD4+ T cells refractory to Treg-mediated suppression in vivo.

## Difference in ribosomal genes between IL-6Rα[T-KO] and WT mice

The gene expression profile of antigen-specific CD4+ T cells of IL-6Rα[T-KO] mice after immunization with protein and LPS is different from the profile of wild-type T cells and T cells from MyD88[T-KO] mice, whose phenotype is similar to that of IL6-Rα[T-KO] mice (*Schenten et al., 2014*). To gain further insights into the IL-6-induced changes in antigen-specific CD4+ T cells, we performed a gene set enrichment analysis (GSEA) of the differentially expressed genes of the cells from IL-6Rα[T-KO] mice and wild-type controls on day 7 after immunization (*Figure 6—figure supplement 2*). Genes whose induction has been associated with the STAT3 signaling pathway were repressed in IL-6Rα[T-KO] mice, thus validating the overall strategy for the analysis. Interestingly, IL-6Rα-deficient CD4+ T cells also upregulated transcripts for cell cycle genes. However, we found the most striking difference between IL-6Rα-deficient CD4+ T cells and wild-type controls in ribosomal genes. A large number of these genes were down-regulated in IL-6Rα-deficient CD4+ T cells, suggesting that protein synthesis is impaired in IL-6Rα-deficient CD4+ T cells, which presumably negatively affects proliferation and cytokine secretion in these cells.

## Discussion

IL-6 is one of the most pleiotropic cytokines that acts on multiple cell types and regulates many aspects of innate and adaptive immunity (*Kamimura et al., 2003*). Consequently, the role of IL-6 signaling specifically in T cells has been very difficult to evaluate. In this study, we focused on a specific question about IL-6 signaling, namely its role in the connection between TLR activation and the generation of a CD4+ T cell response. Using mice bearing a T cell-specific deletion of the IL-6Rα, we found that the absence of IL-6 signaling in T cells resulted in abrogated expansion of CD4+ T cell and impaired development of Th1 and, as expected, Th17 cells following immunization with TLR agonists as adjuvants. In the steady state, IL-6RαT-KO mice had similar numbers of CD4+ and CD8+ T cells as their WT counterparts and we did not observe an increase in the fraction of apoptotic T cells in the thymus or periphery of these mice. These results indicate that the impairment of CD4+ T cell responses in IL-6RαT-KO mice was not due to a T cell-intrinsic defect in survival in the absence of IL-6 signaling. Because we had previously demonstrated that cytokines produced by DCs in response to TLR activation, in particular IL-6, render CD4+ T cells refractory to Treg-mediated suppression in vitro, we hypothesized that the defect in T cell responses in these mice might be caused by unopposed suppression by Tregs. Indeed, depletion of CD25+ Tregs restored the development of the Th1 response in IL-6RαT-KO mice, while the Th17 response was not rescued, which is consistent with the role of IL-6 in the differentiation of Th17 cells. Moreover, deleting the IL-6Rα specifically in Foxp3+ Tregs revealed that IL-6 signaling is not required in these cells for the generation of CD4+ T cell responses and instead suggested that IL-6 signaling in conventional CD4+ T cells was critical for the generation of a CD4+ T cell response. Collectively, these data therefore support a model in which IL-6 signaling in naive T cells makes them refractory to the suppressive activity of Tregs in vivo.

We found that Tfh cells were generated following immunization of IL-6Rα T-KO mice, albeit at moderately reduced frequencies, suggesting that IL-6 is not essential as an instructive signal for the induction of this T cell lineage. STAT3 has been demonstrated to be important for Tfh cell differentiation after protein immunization (*Nurieva et al., 2008*). Thus, in the absence of IL-6 signaling, another STAT3-dependent cytokine such as IL-21 might compensate for IL-6 in inducing the development of Tfh cells. Indeed, IL-21 has been demonstrated to play an important role in Tfh cell generation (*Nurieva et al., 2008*). Tfh cells play a critical role in providing help for GC B cells. Therefore, to examine the effects of IL-6 on Tfh cell function, we analyzed GC formation, plasma cell differentiation and antibody production in IL-6RαT-KO mice. Although Tfh cells could develop in these mice, there was a reduction in the GC compartment size the percentage of plasma cells and, importantly, the amount of antigen-specific antibodies following immunization. Defective IL-6 signaling in CD4+ T cells affected predominantly the titers of antigen-specific IgG2c, while the titers of antigen-specific IgG1 were only modestly reduced. In this context, it is interesting to note that IFN-γ, the signature cytokine of Th1 cells, is important for class-switch recombination (CSR) to IgG2c. The role of Tfh cells in the initiation of CSR has not been fully understood. Under conditions of Th2 immunity, Tfh cells have been identified as the major source of the IL-4 and IFN-γ required for the induction of CSR to IgG1 and IgG2c, respectively (*Veldhoen et al., 2006*). In analogy, defective IFN-γ secretion by Tfh cells in IL-6RαT-KO mice may cause the impairment of the IgG2c response in these mice. Indeed, we found substantial amounts of Tfh cells in the T cell assays with which we measured IFN-γ secretion, which would be consistent with this view (data not shown). Alternatively, it is possible that under the conditions used in our study, CSR is already initiated outside of the GCs and the absence of a functional Th1 response negatively influences CSR to IgG2c. Regardless of these possibilities, however, our data collectively demonstrate that IL-6, while dispensable for the generation of Tfh cells, plays a nonredundant role in the function of these cells. Interestingly, depletion of Tregs recovered the antigen-specific IgG2c titers in IL-6αT-KO mice, suggesting that the activity of Tfh cells is negatively regulated by Tregs and this negative control is opposed by T cell-intrinsic IL-6 signaling.

Tregs have been demonstrated to suppress T cell responses by a variety of mechanisms including the sequestration of IL-2 by Tregs or the production of inhibitory cytokines such as TGF-β, IL-10, and IL-35. In search of the mechanism by which IL-6 counteracts the suppressive activity of Tregs, we found that both Tregs and TGF-β inhibited expression of CD25 on CD4+ T cells and this inhibition could be partially prevented by IL-6 and completely prevented by the combined effect of IL-6 and IL-1. Studies have shown that IL-1 is important for Th17 cell differentiation (*Acosta-Rodriguez et al., 2007*; *Hu et al., 2011*; *Sutton et al., 2006*; *Volpe et al., 2008*). IL-1 has been demonstrated to augment Th17 differentiation induced by IL-6 in combination with TGF-β (*Veldhoen et al., 2006*). Recently,

it has been suggested that IL-6 induces the upregulation of the IL-1R and that this effect is a prerequisite for the differentiation of Th17 cells (*Chung et al., 2009*). Consistent with this view, our own gene expression studies on antigen-specific CD4+ T cells from immunized IL-6RαT-KO and MyD88T-KO mice also revealed a downregulation of the IL-1R in antigen-specific CD4+ T cells in the absence of IL-6 signaling (*Schenten et al., 2014*). While further studies are required to determine whether this particular mechanism also applies to other T cell lineages, it appears that IL-6 and IL-1 cooperate not only in the induction of Th17 differentiation but also in the generation of a Th1 response. In the context of the latter response, our work demonstrates that IL-6 and IL-1 are both important to overcome Treg-mediated suppression, perhaps in part by cooperating in the maintenance of the sensitivity of CD4+ responder T cells to IL-2 and other relevant instructive cytokines such as IL-12 (*Schenten et al., 2014*). It is likely that other cytokines or cytokine combinations may play a similar role in other types of T cell responses, depending on the pathogen and PRRs involved. It should be noted in this regard that several cytokines, including IL-15, have been shown to render CD4+ T cells resistant to the suppressive activity of Tregs in vitro (*Ben Ahmed et al., 2009*). Interestingly, we found that similar to IL-6 and IL-1, IL-15 is also able to counter Treg-mediated downregulation of CD25 in CD4+ T cells (*Figure 6—figure supplement 1*). This raises the possibility that control of CD25 expression on naive CD4+ T cells may be a common mechanism utilized by different cytokines induced by different pathways of innate immune recognition. It was previously demonstrated that IL-15 overcomes Treg-mediated suppression by activating phosphatidylinositol 3-kinases (PI-3 kinases), which are key regulators of cell growth, survival, and proliferation (*Ben Ahmed et al., 2009*). Interestingly, both IL-6 and IL-1 have also been implicated in the initiation of PI-3 kinase signaling (*Hideshima et al., 2001*; *Madge and Pober, 2000*). This raises the possibility that the similarity in the ability of these cytokines to maintain the sensitivity of responder T cells to IL-2 signals in the presence of Tregs may be due to their shared function in inducing this important signaling pathway. Further studies are needed to determine whether inhibiting the PI-3 kinase pathway downstream of IL-6 and IL-1 signaling results in a failure to rescue the down-regulation of CD25 expression and susceptibility to Treg-mediated suppression. It should also be noted that while control of CD25 expression may be applicable to naive CD4+ T cells and the Th1 differentiation pathway, other stages of T cell responses (e.g., memory response) and other differentiation pathways (e.g., Th17 and Tfh) are likely to be counter-regulated by IL-6 and Tregs through a different control point. The counter-regulation by IL-6 and Tregs in these cases may still rely on the same principle, that is, regulated expression of a receptor for an autocrine cytokine that controls cell expansion and maintenance. Possible candidates include IL-21 and the IL-21R (for Tfh and Th17 cells), and IL-4 and the IL-4 receptor for Th2 cells. Indeed, our in vitro studies indicate that IL-6 and Tregs may also counter-regulate the IL-4Rα, IL-21R, and the common γ-chain (data not shown). Future studies will need to address this possibility in vivo.

We investigated the changes in the cellular pathways in IL-6Rα-deficient CD4+ T cells by GSEA of antigen-specific CD4+ T cells on day 7 after immunization. The late time point allowed us to obtain enough cells for the analysis but likely missed important changes in the gene expression at early stages of the immune response. Thus, the observed changes in the cellular pathways in IL-6Rα-deficient CD4+ T might be biased by the presence of IL-6Rα-deficient CD4+ T cells that have escaped the Treg-mediated control mechanisms. This caveat of the analysis aside, we did observe a repression of STAT3-induced genes, thereby confirming the general approach of the analysis. IL-6Rα-deficient CD4+ T cells up-regulated cell cycle genes during the late stage of the immune response. At this point, the cell may attempt to proliferate, particularly in light of the lower absolute numbers of antigen-specific CD4+ T cells from IL-6RαT-KO mice, but are unable to do so because of a failure to induce ribosome production. The latter observation, namely a defect in the up-regulation of ribosome-associated genes in IL-6Rα-deficient CD4+ T cells, appeared to be one of the most significant changes in these cells. Interestingly, CD4+ T cells heterozygous for the ribosomal protein S6, which is also down-regulated in IL-6Rα-deficient CD4+ T cells, increase their size after TCR stimulation but fail to proliferate (*Sulic et al., 2005*). Indeed, increased ribosome production accompanies increased cytokine production following TCR stimulation (*Asmal et al., 2003*). Thus, decreased ribosome generation may be a consequence of Treg-mediated suppression in the absence of IL-6 signal. Further studies will be required to elucidate this aspect.

After microbial infection, a population of memory precursor cells are generated that progressively differentiate into functionally mature, long-lived memory T cells (*Cui and Kaech, 2010*). It is currently unknown whether particular cytokines produced during infection are required for the differentiation and maintenance of memory CD4+ T cells. Previously, using MyD88-deficient mice, we showed that a

TLR-induced signal was necessary for the formation of memory CD4[+] T cell responses (*Pasare and Medzhitov, 2004*). However, the nature of the signal was unclear at the time. In the present study, we found that although depleting Tregs rescued the primary Th1 response in the absence of IL-6 signaling in T cells, the memory response was still defective. Furthermore, we found that antigen-specific memory CD4[+] T cells were still generated, even though the IFN-γ response was impaired in IL-6Rα[T-KO] mice. This result suggests that IL-6 signaling is required for the generation of functionally competent memory CD4[+] T cells. Importantly, in a parallel study, we found that the functionally competent memory CD4[+] T cells also depend on T cell-specific IL-1 signaling, suggesting that IL-6 and IL-1 cooperate in this aspect of the CD4[+] T cell response as well (*Schenten et al., 2014*). Finally, it is interesting to note that STAT3-dependent cytokines, such as IL-6, IL-10, and IL-21, as well as MyD88-dependent IL-1 have been suggested to play a role in memory CD8[+] T cell differentiation and functional maturation following immunization and infection (*Foulds et al., 2006*; *Hinrichs et al., 2008*; *Castellino and Germain, 2007*; *Cui et al., 2011*; *Yi et al., 2010*). The fact that both IL-6Rα- and MyD88-deficient memory cells persist for several months following the primary immunization suggests that other signals control the maintenance of these cells.

In summary, this study demonstrates that T cell-intrinsic IL-6 signaling plays a critical role in CD4[+] T cell expansion, differentiation, and memory formation. Moreover, we have shown that IL-6 in cooperation with IL-1β counteract the suppressive activity of Tregs by maintaining CD25 expression in CD4[+] T cells. Thus, cytokines induced upon innate immune recognition, including IL-6 and IL-1, can control activation of adaptive immune responses by rendering antigen-specific T cells refractory to suppression by Treg cells. Disregulation of this mechanism may also contribute to the development of autoimmune diseases.

## Materials and methods

### Animals

The generation of the conditional *Il6ra* allele has been published previously (*Wunderlich et al., 2010*). Briefly, exon 2 and 3 were flanked by *loxP* sites. Deletion of these exons upon Cre-mediated recombination leads to a frameshift mutation and a premature stop codon in exon 4. As a result, only exon 1 and an additional non-sense 20 aa are encoded by the deleted allele, resulting in a non-functional protein without a binding site to IL-6. *Cd4-Cre* (*Lee et al., 2001*), *Foxp3-Cre* (*Rubtsov et al., 2008*), and *Foxp3-GFP[KI]* mice (*Fontenot et al., 2005*) have been previously described. All animals were kept on a C57BL/6 background. Animals were housed in a conventional, specific pathogen-free facility at Yale University and all animal experiments were performed in accordance with the guidelines set by the Institutional Animal Care and Use Committee of Yale University.

### Reagents and antibodies

Ovalbumin, LPS, Complete Freund's Adjuvant (CFA) and Incomplete Freund's Adjuvant (IFA) were purchased from Sigma Aldrich (St. Louis, MO). Endotoxin-free OVA was obtained from Biovendor, LLC (Candler, NC). Low endotoxin CpG and peptidoglycan (PGN) were purchased from Invivogen (San Diego CA). MOG$_{35-55}$ peptide was purchased from Anaspec and 2WS1 peptide (EAWGALANWAVDSA) was from Genscript (Piscataway, NJ). Pertussis toxin was purchased from Calbiochem/EMD Millipore (Billerica, MA). Antibodies used included: CD4, CD8, CD126, CD25, CD44, CD19, B220, IFN-γ, IL-17, CXCR5, PD-1, CD138, CD95, CD45Rb, Active Caspase-3, CD62L, pSTAT3 (Y705), ICOS, PSGL1, CD3ε, CD28, IgG1 and IgG2c (all purchased from BD biosciences, San Diego, CA) and PNA (Vector Laboratories, Burlingame, CA). Recombinant IL-6, IL-1β, and IL-15 were purchased from R&D Systems (Minneapolis, MN). Annexin-V and the Foxp3 staining kit were purchased from Ebioscience (San Diego, CA). Anti-CD4 microbeads were purchased from Miltenyi Biotec (Auburn, CA). Mouse cells were cultured in complete RPMI-1640 supplemented with 10% FCS, 2 mM L-glutamine, 1 mM Sodium pyruvate, 50 μM β-mercaptoethanol, 10 mM Hepes, 100 U/ml Penicillin, 100 μg/ml Streptomycin, all from Gibco/Life Technologies (Grand Island, NY). 2W:I-A[b] tetramers were a generous gift from Marc Jenkins (University of Minnesota, Minneapolis, Minnesota).

### Immunizations and Treg depletion

Mice were injected subcutaneously in both hind footpads with either 50 μg/footpad of OVA or 50 μg/footpad of 2WS1 peptide (EAWGALANWAVDSA; Genscript) along with 5 μg/footpad LPS emulsified in IFA. For certain experiments, mice were immunized with OVA and CpG or PGN emulsified in IFA or OVA emulsified in CFA. For memory experiments, mice were immunized in only one hind footpad for the primary immunization and 60 days later a secondary immunization was done in the opposite hind

footpad. For in vivo depletion of CD4+ CD25+ Tregs, mice received an intravenous injection of 100 µg of monoclonal α-CD25 antibody (clone PC61). 3 days later, depletion of Tregs was confirmed by staining peripheral blood lymphocytes for CD4 and CD25 markers.

## Surface and intracellular staining

Cells were stained with relevant antibodies for 30 min on ice for cell surface staining. For 2W:IAb tetramer staining, cells were stained for 1 hr at room temperature. For intracellular staining of cytokines, cells were stimulated in the presence of Golgiplug (BD Biosciences) for 5 hr, stained for cell surface molecules, fixed, permeabilized, and stained for intracellular cytokines using the BD Biosciences intracellular cytokine staining kit. For detecting Foxp3 expression, cells were stained using the BD Biosciences Foxp3 staining kit. For pSTAT3 staining, cells were stimulated with IL-6 in vitro for 20 min, fixed and permeabilized with 90% methanol, then stained with anti-pSTAT3 (Y705) antibody. Cells were analyzed on a FACSCalibur flow cytometer or LSRII (BD Biosciences) with FlowJo software (Tree Star, Ashland, OR).

## T cell purification and in vitro restimulation

Popliteal and inguinal lymph nodes were isolated from mice and single-cell suspensions were incubated with anti-CD4 microbeads (Miltenyi Biotec) and subsequently MACS-purified. T cell purity was confirmed by flow cytometry. Purified CD4+ T cells ($1 \times 10^5$) were cultured in U-bottom 96-well tissue culture plates with $3 \times 10^5$ irradiated splenocytes as antigen-presenting cells (APCs) and titrating doses of antigen for 72–84 hr. To assess proliferation of T cells, [³H]-thymidine was added for the last 12–16 hr of the culture. Supernatants were collected at approximately 84 hr to determine cytokine production by ELISA.

## Western blot

CD4+ T cells were MACS-purified using anti-CD4 beads and stimulated with mAbs against CD3e and CD28 for 15 min. When indicated, the cells were stimulated in the presence of 20 ng/ml IL-6 (R&D Systems), 20 ng/ml IL-6 + 400 ng/ml sIL-6Ra (Santa Cruz, Germany), 20 ng/ml IL-11 (Peprotec, Germany), 20 ng/ml OSM (Sigma), or 20 ng/ml CNTF (R&D Systems). Phosphorylation of STAT3 was detected by Western blot using an anti-pSTAT3 mAb (Cell Signaling, Germany). Equal loading was ensured by staining for β-actin (Santa Cruz).

## Quantitative PCR

Following immunization, RNA was isolated from PD-1+ CXCR5+ CD4+ Tfh and PD1− CXCR5− CD4+ non-Tfh cells sorted by flow cytometry, three mice per genotype, using the microRNeasy Kit (Qiagen, Valencia, CA). Quantitative PCR was performed using the following primers: *bcl6*, 5'-CACACTCGAATTCACTCTG-3' (forward) and 5'-TATTGCACCTTGGTGTTGG-3' (reverse) and *il21*, 5'-AGGGCCAGATCGCCTCCTGATT-3' (forward) and 5'-GAGCTGGGCCACGAGGTCAATG-3' (reverse).

## Immunofluorescence

Popliteal and inguinal lymph nodes isolated from mice 14 days following immunization were immediately frozen in OCT tissue-freezing medium. Sections were cut to 6-µm thickness on a cryostat and fixed in acetone. Adjacent sections were stained with B220-FITC and PNA-biotin followed by Streptavidin-APC or B220-FITC and CD4-APC, respectively.

## Enzyme linked immunosorbent assay (ELISA)

Paired antibodies against IFN-γ, IL-17A, IgG1, and IgG2c were purchased from BD biosciences to perform ELISAs.

## Treg suppression and cytokine stimulation assay

CD4+ Foxp3− and CD4+ Foxp3+ cells were sorted from the spleens of Foxp3-GFP mice by flow cytometry. Purified CD4+ Foxp3− T cells ($5 \times 10^4$) were cultured in U-bottom 96-well tissue culture plates with CD4+ Foxp3+ T cells ($5 \times 10^4$) and soluble α−CD3ε (4 µg/ml) and α−CD28 (4 µg/ml) plus or minus stimulation with recombinant cytokines: IL-6 (100 ng/ml), IL-1β (100 ng/ml) and IL-15 (100 ng/ml). Alternatively, CD4+Foxp3− T cells were stimulated in flat-bottom 96-well tissue culture plates with plate-bound α-CD3ε (1 µg/ml) and α-CD28 (1 µg/ml) and TGF-β (10 ng/ml) was added to the cultures instead of CD4+Foxp3+ cells. While we kept the concentration of IL-6 constant, we noticed a loss of IL-6 bioactivity upon reconstitution in PBS containing BSA and storage at −80°C. Fresh IL-6 was largely sufficient to maintain CD25 expression on CD4+ T cell in the presence of Tregs, whereas limiting amounts of IL-6 necessitated the addition of IL-1β to reach the same effect.

## Gene set enrichment analysis

Gene set enrichment analysis (GSEA) is a computational tool for determining the enrichment of previously characterized gene sets at the top or bottom of a rank ordered list. The generation of the original data describing the general gene expression profile of antigen-specific CD4[+] T cells from IL-6Rα[T-KO] mice and wild-type control has been published elsewhere (*Schenten et al., 2014*). For the GSEA, the rank-ordered-list was created by calculating the fold change between the FPKM of antigen-specific CD4[+] T cells from IL-6Rα[T-KO] and wild-type mice and subsequently ordering the list by the fold change. FPKM values of the CD4[+] T cells from two mouse strains were determined using the Cuff-diff program. A pseudocount of 0.01 was added to all FPKM values to prevent dividing by zero and only genes with an FPKM ≥1 in either group were included in the analysis. The GseaPreranked tool, part of the javaGSEA Desktop Application v2.0.14, was used in conjunction with the Molecular Signatures Database v4.0 to run the analysis.

## EAE induction

EAE was induced by subcutaneous immunization of mice in the rear flank region with 200 µl of an emulsion of 300 µg of $MOG_{35-55}$ peptide and 250 µg of M. tuberculosis H37RA in CFA on days 0 and 7. Mice also received two injections of 500 ng of pertussis toxin in 200 µl total volume intraperitoneally on days 0 and 2. Clinical signs of EAE were assessed according to the following score: 0, no sign of disease; 1, loss of tone in tail; 2, paraparesis; 3, hind limb paralysis; 4, quadraplegia; 5, moribund. Mice were euthanized before they reached stage 5.

## Colitis induction

CD4[+] CD45RB[hi] cells were sorted from the spleen of WT mice by flow cytometry and $5 \times 10^5$ cells were adoptively transferred intraperitoneally into recipient RAG2 KO mice. Mice were monitored for disease progression by weighing each mouse weekly using a top-loading balance. Mice typically started developing signs of disease 4–5 weeks after the transfer of CD4[+] CD45RB[hi] T cells.

## Statistical analysis

Where indicated, p values for statistical significance were determined by either two-tailed unpaired Student's *t* test or Mann–Whitney test. Number of asterisks represents the extent of significance with respect to p value.

## Acknowledgements

We would like to thank Jorge Henao-Mejia, Charles Annicelli, and Sophie Cronin for technical assistance; Marc Jenkins for reagents; Alexander Rudensky for mice; and Noah Palm and Jelena Bezbradica for insightful discussions and critical reading of the manuscript. This work was supported by the National Institutes of Health (RO1 AI055502, AI046688, AI089771 and DK071754 to RM), the Howard Hughes Medical Institute (RM), the CMMC, the Novartis Foundation, and the German Research Foundation (grants 1492-7-1 and SFB 635 to JCB and SFB 832 to JCB and FTW). SAN was supported by an NIH training grant and DS received an NIH training grant and an Irvington Fellowship of the Cancer Research Institute.

## Additional information

### Competing interests

RM: Reviewing editor, *eLife*. The other authors declare that no competing interests exist.

### Funding

| Funder | Grant reference number | Author |
| --- | --- | --- |
| National Institutes of Health | AI046688 | Ruslan Medzhitov |
| National Institutes of Health | AI055502 | Ruslan Medzhitov |
| National Institutes of Health | DK071754 | Ruslan Medzhitov |
| Howard Hughes Medical Institute | N/A | Ruslan Medzhitov |

The funder had no role in study design, data collection and interpretation, or the decision to submit the work for publication.

## Author contributions

SAN, DS, Conception and design, Acquisition of data, Analysis and interpretation of data, Drafting or revising the article; FTW, Analysis and interpretation of data, Contributed unpublished essential data or reagents; SDP, Data analysis, Analysis and interpretation of data, Drafting or revising the article; YG, NH, SY, HKL, LP, IB, BY, Acquisition of data, Analysis and interpretation of data; XY, HZ, Analysis and interpretation of data, Drafting or revising the article; JB, Drafting or revising the article, Contributed unpublished essential data or reagents; RM, Conception and design, Analysis and interpretation of data, Drafting or revising the article

## Ethics

Animal experimentation: This study was performed in accordance with the recommendations in the Guide for the Care and Use of Laboratory Animals of the National Institutes of Health. All of the animals were handled according to approved institutional animal care and use committee (IACUC) protocols (#2011-08006) of the Yale University.

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
