## [Decision Letter]

Thank you for sending your work entitled “T cell-intrinsic role of IL-6 signaling in primary and memory responses” for consideration at *eLife*. Your article has been evaluated by a Senior editor and two peer reviewers, one of whom is a member of our Board of Reviewing Editors.

While we find your work of potential interest, the reviewers have raised substantial concerns, as you find below, that in our view need to be addressed before we can consider publication in *eLife*. Should further experimental data allow you to address these criticisms, we would be happy to look at a revised manuscript.

We have extracted the main points to address here:

1) In many of the in vivo studies it is not clear whether the actions of IL-6 signaling in T cells are cell intrinsic functions. It would be fairly straightforward to make bone marrow chimeras with WT and IL-6RKO bone marrow and track the fate of IL-6RKo T cells in terms of primary and secondary responses and sensitivity to Treg mediated suppression. Some information on proliferation versus cell survival pathways in the various T cell functions analysed would increase mechanistic insight.

2) The authors nicely complement their previous studies showing that IL-6 overcomes the suppressive function of Treg cells. They present evidence in vitro for impaired cytokine receptor expression. However it is widely appreciated that an vitro suppression assay bears little resemblance to T-reg mediated suppression in vivo. Some further validation of the in vitro results in vivo would increase impact of the findings.

3) Figure 1—figure supplement 2. The authors conclude that impaired colitis following transfer of IL-6Rako T cells is due to diminished Th17 differentiation. Nuerath and colleagues showed in 2000 (Nat Med) that lack of IL-6 in this model leads to T cell apoptosis. This may explain the results presented here and the authors should measure T cell accumulation following transfer as well as cytokine production to support their conclusions.

4) In most of the studies, CD4^+^ T cell proliferation and cytokine production were examined in mixed populations (in the presence of CD4^+^ Treg cells). So, just wondering whether the intrinsic proliferation and cytokine production of specific CD4^+^ T cell populations were impaired once IL-6 signaling is abrogated? Especially in Figure 1, for the proliferation of CD4^+^ T cells after in vitro restimulation assays, were the irradiated splenocytes derived from wild type mice? It may be better to use wild type antigen-presenting cells (e.g., DC) to replace splenocytes since these cells derived from WT or IL-6Rα KO mice may be different.

5) Similarly, in Figure 1—figure supplement 3, FACS analysis of IFNγ and IL-17 may be required for understanding the effects of IL-6 signaling on the specific T cell populations.

6) In Figure 2, the authors used Treg depletion to illustrate IL-6 effects. However, as shown in Figure 2—figure supplement 3, Treg cells were also affected by IL-6Rα deficiency. Whether IL-6Rα deficiency affects the suppressive functions of Treg cells/per cell should be clarified in addition to the analysis of Treg numbers.

7) Figure 6—figure supplement 1, IL-15 could also overcome inhibitory effects of Treg cells. Since IL-15 and IL-6 signals via different signaling pathway, the authors may need to discuss the potential molecular mechanisms involved in this phenomenon.

8) IL-6Rα-deficient mice were used in this study. However, one possibility that needs to be excluded is the off-targets effects of KO. Did the authors examine the rescue effects of IL-6Rα or gp130 signaling on CD4^+^ T cell activation and differentiation? The authors should discuss this issue.

9) In Figure 1, p-STAT3 is examined after IL-6 treatments. It may be needed to examine the integrity of gp130 signaling axis, e.g., in response to IL-6-like cytokines.

10) Although reference was cited for IL-6Rα deficient mice, it may be better to describe the KO effects on IL-6Rα (complete deletion of ORF or truncated IL-6Rα) since this information is very important for evaluation of this study.

---

## [Author Response]

*1) In many of the in vivo studies it is not clear whether the actions of IL-6 signaling in T cells are cell intrinsic functions. It would be fairly straightforward to make bone marrow chimeras with WT and IL-6RKO bone marrow and track the fate of IL-6RKo T cells in terms of primary and secondary responses and sensitivity to Treg mediated suppression. Some information on proliferation versus cell survival pathways in the various T cell functions analysed would increase mechanistic insight*.

This aspect may have been overlooked during the initial review of our manuscript, but we would like to highlight that we have used conditional KO mice that carry a T cell-specific ablation of the IL-6Rα throughout our study. All actions of IL-6 must be therefore T cell-intrinsic. We have added more information in Figure 1—figure supplement 6 on the ability of antigen-specific T cells to proliferate or survive.

*2) The authors nicely complement their previous studies showing that IL-6 overcomes the suppressive function of Treg cells. They present evidence in vitro for impaired cytokine receptor expression. However it is widely appreciated that an vitro suppression assay bears little resemblance to T-reg mediated suppression in vivo. Some further validation of the in vitro results in vivo would increase impact of the findings*.

We agree that it would be desirable to provide more insights into the molecular mechanisms that allow CD4^+^ T cells to escape Treg-mediated suppression in vivo. However, several technical hurdles make this effort challenging. First, CD25 expression after the activation of CD4^+^ T cells is known to be very transient and we are unable to detect enough antigen-specific CD4^+^ T cells with an activated phenotype (CD44^+^) at early time points (<3 days) of the immune response. Second, the analysis of antigen-specific CD4^+^ T cells at later stages of the response (day 4-7) is hampered by the fact that one presumably only interrogates cells in IL-6Rα^T-KO^ mice that have escaped Treg-mediated suppression. Thus, the analysis might be biased by the presence of these escapees during the late stages of the response. This caveat aside, we have performed a Gene Set Enrichment Analysis (GSEA) on the gene expression profiles of antigen-specific CD4^+^ T cells from IL-6Rα^T-KO^ mice and wild-type controls in order to gain more mechanistic insights. In the new Figure 6—figure supplement 2, we present the interesting finding from this analysis that IL6Ra-deficient CD4^+^ T cells are severely impaired in their ribosomal metabolism. The implications of this finding are evaluated in the expanded Discussion section.

*3)*
Figure 1—figure supplement 2*. The authors conclude that impaired colitis following transfer of IL-6Rako T cells is due to diminished Th17 differentiation. Nuerath and colleagues showed in 2000 (Nat Med) that lack of IL-6 in this model leads to T cell apoptosis. This may explain the results presented here and the authors should measure T cell accumulation following transfer as well as cytokine production to support their conclusions*.

The referees raise an interesting and important mechanistic point about the role of IL-6 in Th17 differentiation during colitis. We would like to emphasize that we used the T cell transfer model of colitis in our study mainly to demonstrate that our experimental system, namely the use of IL-6Rα^T-KO^ mice, behaves as expected with respect to the differentiation of Th17 cells. The role of IL-6 in Th17 differentiation is not the focus of our study and the suggested experiments might distract from the core message of our manuscript. That being said, we show in the new Figure 1—figure supplement 6 that at least in the context of immunizations, IL-6Rα-deficient antigen-specific CD4^+^ T cells are not more prone to undergo apoptosis.

*4) In most of the studies, CD4*^*+*^
*T cell proliferation and cytokine production were examined in mixed populations (in the presence of CD4*^*+*^
*Treg cells). So, just wondering whether the intrinsic proliferation and cytokine production of specific CD4*^*+*^
*T cell populations were impaired once IL-6 signaling is abrogated? Especially in*
Figure 1*, for the proliferation of CD4*^*+*^
*T cells after in vitro restimulation assays, were the irradiated splenocytes derived from wild type mice? It may be better to use wild type antigen-presenting cells (e.g., DC) to replace splenocytes since these cells derived from WT or IL-6Rα KO mice may be different*.

The main concern of the referees, namely that the composition or function of the irradiated splenocytes used in the in vitro re-stimulation assays may differ between IL-6Rα^T-KO^ mice and wild-type controls and thus affects the outcome of these assays, had already been addressed. In all of these assays, we isolated CD4^+^ T cells from the draining lymph nodes of immunized IL-6Rα^T-KO^ mice and wild-type controls and co-cultured them with irradiated splenocytes from naïve wild-type mice. Thus, differences in the outcome of these assays must reflect the differences in CD4^+^ T cells, which also are the only cells capable of proliferation. Defective IL-6 signaling in Tregs cannot account for these differences as Tregs are not present in concentrations that would typically result in the suppression of responder T cells in vitro. Moreover, we have shown that IL-6Rα-deficient Tregs suppress as efficiently as wild-type Tregs in vitro (new Figure 2—figure supplement 3). The effects on proliferation and cytokine production of specifc responder T cells (Th1, Th17) are shown in Figure 1.

*5) Similarly, in*
Figure 1—figure supplement 3*, FACS analysis of IFNγ and IL-17 may be required for understanding the effects of IL-6 signaling on the specific T cell populations*.

The purpose of this figure is to show that the initial phenotype of an impaired Th1 and Th17 response in IL-6Rα^T-KO^ mice is not confined to the use of LPS as an adjuvant but also applies to other TLR ligands. For the remainder of the study, we have focused on the response to LPS and show the effects of IL-6 signaling on specific T cell populations via intracellular FACS analysis in Figure 1.

*6) In*
Figure 2*, the authors used Treg depletion to illustrate IL-6 effects. However, as shown in*
Figure 2—figure supplement 3*, Treg cells were also affected by IL-6Rα deficiency. Whether IL-6Rα deficiency affects the suppressive functions of Treg cells/per cell should be clarified in addition to the analysis of Treg numbers*.

Figure 2—figure supplement 3 does indeed show that the absolute number of Tregs is reduced in IL6Rα^T-KO^ mice. However, this reduction can be mainly attributed to the diminished CD4^+^ T cell response in these mice, which is accompanied by a reduced overall size of the draining lymph nodes. The frequency of the Tregs relative the overall CD4^+^ T cell population does not change in IL-6Rα^T-KO^ mice. Moreover, we showed in the original manuscript that a Treg-specific ablation of the IL6Ra has no effect on the immune response. Together, these aspects of the phenotype of IL-6Rα^T-KO^ mice make it unlikely that the suppressive functions of Tregs are impaired in the absence of IL-6. Nonetheless, in order to clarify this aspect, we show in the new Figure 2—figure supplement 3 that IL-6Rα-deficient Tregs are as capable to suppress as wild-type cells.

*7)*
Figure 6—figure supplement 1*, IL-15 could also overcome inhibitory effects of Treg cells. Since IL-15 and IL-6 signals via different signaling pathway, the authors may need to discuss the potential molecular mechanisms involved in this phenomenon*.

We appreciate the suggestion of the referees and we have modified our manuscript to discuss this aspect.

*8) IL-6Rα-deficient mice were used in this study. However, one possibility that needs to be excluded is the off-targets effects of KO. Did the authors examine the rescue effects of IL-6Rα or gp130 signaling on CD4*^*+*^
*T cell activation and differentiation? The authors should discuss this issue*.

The IL-6Rα is specific for IL-6. As no other ligands are known for the IL-6Rα, it is unlikely that off-targets effects are responsible for the observed phenotype of IL-6RαT-KO mice. Nonetheless, in order to address the concerns of the referees, we have examined the ability of IL-6Rα-deficient CD4 T cells to phosphorylate STAT3 upon addition of the soluble form of the IL-6Rα plus IL-6. Naive IL-6Rα-deficient CD4 T cells phosphorylated STAT3 as efficiently as wild-type cells upon addition of soluble IL-6Rα + IL-6, suggesting that the gp130 signaling axis remains intact in IL-6Rα-deficient CD4 T cells (see Figure 1—figure supplement 1).

*9) In*
Figure 1*, p-STAT3 is examined after IL-6 treatments. It may be needed to examine the integrity of gp130 signaling axis, e.g., in response to IL-6-like cytokines*.

We have also stimulated naïve IL-6Rα-deficient CD4 T cells with the gp130-dependent cytokines IL-11, OSM, and CNTF. However, none of these additional cytokines were able to phosphorylate STAT3 efficiently (see Figure 1—figure supplement 1). Thus, other gp130-dependent cytokines presumably do not play a major role during the activation phase of the CD4^+^ T cell response. It remains possible that these cytokines are important to the physiology of T cells at later stages of the effector phase. However, these events were not the focus of our study.

*10) Although reference was cited for IL-6Rα deficient mice, it may be better to describe the KO effects on IL-6Rα (complete deletion of ORF or truncated IL-6Rα) since this information is very important for evaluation of this study*.

For the deletion of the IL-6Rα, exon 2 and 3 were flanked by loxP sites. Deletion of exon 2 and 3 upon Cre-mediated recombination leads to a frameshift mutation and a premature stop codon in exon 4. As a result, only exon 1 and an additional non-sense 20 aa are encoded by the deleted allele. The binding site to IL-6 is encoded by the deleted exon 2 and 3, thus the deleted allele produces a non-functional protein. We have added this information in the manuscript.